# Production of Resveratrol Glucosides and Its Cosmetic Activities

**Samir Bahadur Thapa [1], Juhee Jeon [2] , Byung Gyu Park [2] , Dabin Shim [3], Chang Seok Lee [3] and Jae Kyung Sohng [1,4,*]**

[1] Department of Life Science and Biochemical Engineering, Sun Moon University, Asan-si 31460, Republic of Korea; thapa.samir2011@gmail.com

[2] Activon Co., Ltd., 46-5, Cheongju-si 28104, Republic of Korea; jhjeon2110@activon.kr (J.J.); bgpark1503@activon.kr (B.G.P.)

[3] Department of Beauty and Cosmetic Science, Eulji University, Seongnam-si 13135, Republic of Korea; ejsdb0126@naver.com (D.S.); cslee2010@eulji.ac.kr (C.S.L.)

[4] Department of Pharmaceutical Engineering and Biotechnology, Sun Moon University, Asan-si 31460, Republic of Korea

[*] Correspondence: sohng@sunmoon.ac.kr; Tel.: +82-41-530-2246

**Abstract:** A biocatalytic system that could produce bioactive resveratrol poly-glucosides, using sucrose as a low-cost source of UDP-glucose donors and amylosucrase DgAS from *Deinococcus geothermalis*, was developed in this study. This system boasts several advantages, including the rapid and direct conversion of substrates to products, thermostability, regio-stereospecificity, and effectiveness, both in vitro and in vivo, at 40 °C. The results showed that the optimal reaction condition of the production of resveratrol glucosides was obtained by 2.0 μg/mL DgAS and 100 mM sucrose at pH 7.0, incubated at 40 °C for 5 h. With a success rate of around 97.0% in vitro and 95.0% in vivo in a short period of time, resveratrol-*O*-glucosides showed exciting outcomes in cosmetic applications, including antioxidant, anti-inflammatory, anti-aging, and whitening effects when tested with Raw 264.7, B16, and HS68 cell lines. DgAS is recognized as an important biocatalyst due to its high thermostability, effectiveness, and specificity among all known amylosucrases (ASases) in the production of poly-glucosides in a chain of polyphenols, such as resveratrol, making it an ideal candidate for industrial use in the cost-effective production of cosmetic items.

**Keywords:** transglucosylation; amylosucrase; polymerization; regio-stereospecific

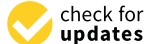



## 1. Introduction

Resveratrol is a polyphenol synthesized by plants as a secondary metabolite to protect them from environmental stresses and microbial infections [1,2]. Chemically, it exists in both *trans-* and *cis-*forms due to a double styrene bond between two phenolic rings [3]. Resveratrol has gained popularity in health-related products, dietary supplements, cosmetics, and medicines due to its multiple beneficial properties for human health, such as cardiovascular protection, as well as anti-aging, anti-cancer, and antioxidant effects [3].

Resveratrol has been reported to be able to eliminate free radicals, such as reactive nitrogen species (RNS) and reactive oxygen species (ROS), that can cause damage to sensitive organs and lead to lipid peroxidation. Resveratrol can enhance cholesterol efflux in the blood, improves cardiovascular and neurological activities, and displays anticancer activities by regulating the expression of small RNA and enhancing transcription factors [4–6]. Additionally, resveratrol is effective in treating obesity, type-II diabetes, Alzheimer's disease, atherosclerosis, ischemic injury, hypertension, and various cancers [7,8]. However, its poor solubility in an aqueous medium remains a challenge for its effective utilization [9,10]. Therefore, different methods have been applied to overcome this limitation, including the use of analog natural products [11].

Resveratrol is very sensitive to pH, light, and high temperatures due to its double C-C bond and unstable hydroxyl groups. Therefore, many studies have been carried out regarding the stability of resveratrol in an effort to make its application more extensive. At room or body temperature, *trans*-resveratrol is stable under low pH. However, it degrades quickly in alkaline conditions [12,13]. Some techniques such as the use of zein–pectin core/shell nanoparticles, the co-encapsulation of α-tocopherol, the inclusion of PEG-modified liposomes, and modification with cyclodextrin can be used to successfully enhance the stability of resveratrol [14–17]. Recent studies have focused on the cosmetic application of resveratrol, as it has shown many positive cosmetic values, in addition to its nutraceutical and pharmaceutical properties [3,7]. Quality cosmeceutical products are in high demand around the world. Many cosmetic industries are working to meet the demand for quality make-up items. As one popular chemical, ascorbic acid (AA) is commonly included in anti-aging and beauty creams due to its efficacy and essential nature. However, AA has limitations, since it is water-soluble, weak, and very unstable. It easily deteriorates or oxidizes when exposed to light, air, high temperatures, alkali, copper, and heavy metals. Ravetti et al. and Boo et al. have found that glycosylation at the 2-position of ascorbic acid (2-glucosyl ascorbic acid) can increase its stability against oxidation and reduction [18,19]. Similarly, some researchers have focused on the enzymatic production of 2-glucosyl glycerol (2GG), as it is a highly expensive, stable, soluble, and valuable cosmetic ingredient included in many beauty creams [20,21].

Glycosylation is a popular method for modifying natural products, including plant polyphenols such as flavonoids, to enhance their pharmacological properties, i.e., bioavailability, solubility, activity, and stability [22]. However, the chemical synthesis of glycosylated flavonoids involves challenges such as strict reaction conditions, protection and deprotection groups, low stereospecificity and regioselectivity, and difficulty in purification, resulting in high production costs and time [23]. To overcome these obstacles, microbial cell factories have been engineered with appropriate sugar biosynthetic pathway genes and glycosyltransferases (GTs) from heterologous sources for the production of targeted glycosylated flavonoids [24,25]. However, high concentrations of some metabolites might be toxic to engineered cells [26,27].

An alternative to microbial cell factories is in vitro enzymatic glycosylation using amylosucrase ASase (EC 2.4.1.4), a versatile enzyme exhibiting hydrolysis, isomerization, and transglycosylation activities [28]. Amylosucrase from different sources has been used to transglycosylate different types of polyphenols, including flavonoids [28–30]. However, the glycodiversification of flavonoids using amylosucrase is challenging, despite attempts to engineer these enzymes to improve their tolerance to various sucrose analogs [31]. In this study, we used DgAS, an amylosucrase obtained from *Deinococcus geothermalis* DSM 11300, for the direct transglycosylation of resveratrol [30]. Furthermore, immobilizing the DgAS enzyme not only drastically reduces the cost of enzyme production and complicated downstream processing but also helps to improve temperature stability, enzyme activity, and selectivity. Various techniques are available for protein immobilization, including cross-linking, adsorption, covalent attachment onto solid supports, and entrapment [32,33].

From an enzyme mechanism perspective, amylose polymerization and transglycosylation employ the same mechanism. The only difference lies in the specificity of the acceptor substrate. ASase (AS) exhibits a broad acceptor substrate specificity. It catalyzes the breakdown of the glycosidic bond in sucrose (α-D-glucopyranosyl-(1→2)-β-fructofuranoside), resulting in the release of fructose and a glucosyl-AS intermediate product [34]. Subsequently, if the glucosyl-AS intermediate attaches to the acceptor molecule, such as maltooligosaccharide (Gn) or the hydroquinone form of polyphenolic compounds, at the +1 subsite instead of to water, it forms a bond with the acceptor substrate in its α-anomer form. This type of reaction is known as transglycosylation. Similarly, the fructose released initially can be used as an acceptor substrate by AS to produce turanose and trehalulose through the isomerization reaction [34,35].

In this study, we employ a DgAS, amylosucrase, to produce resveratrol glucosides more effectively, easily, rapidly, using environmentally friendly in vivo and in vitro systems, employing sucrose as a cheap source of glucose (Figure 1) [36,37]. We focused on evaluating a wide range of cosmetic properties of resveratrol-*O*-glucosides using HS68, B16, and Raw264.7 cell lines for their potential industrial application.

**Figure 1.** Representative reaction scheme of resveratrol glucoside formation by the enzyme DgAS, amylosucrase.

## 2. Materials and Methods

### 2.1. Chemicals and Reagents

Resveratrol, starch, maltodextrin, cyclodextrin glucanotransferase (CGTase, from *Bacillus* sp.), and amyloglucosidase (from *Aspergillus niger*) were purchased from Sigma-Aldrich (St. Louis, MO, USA). Sucrose was purchased from MBcell (Seoul, Republic of Korea). Luria–Bertani (LB) broth medium was obtained from KinsanBio (Seoul, Republic of Korea). All other chemicals and required reagents were of the highest grade and were purchased from different sources. High-performance liquid chromatography (HPLC)-grade acetonitrile and water were purchased from Mallinckrodt Baker (Phillipsburg, NJ, USA). All restriction enzymes were purchased from Takara Bio (Nojihigashi, Kusatsu, Shiga, Japan) and Promega (Madison, WI, USA).

### 2.2. Bacterial Strains, Cloning, and Culture Conditions

For the expression and production of DgAS in pHCE IIB (NdeI), *Escherichia coli* BL21 (DE3) (Stratagene, La Jolla, CA, USA) was used. For all DNA manipulation, *E. coli* XL1 blue was used as a host. Recombinant *E. coli* strains were grown in LB broth or an agar plate supplemented with ampicillin (100 μg/mL) at 37 °C. The recombinant strain *E. coli* (DE3) harboring pHCE IIB (NdeI)-DgAS was prepared by heat shock transformation. The nucleotide sequence of amylosucrase (GenBank Accession No. ABF44874.1) obtained from *Deinococcus geothermalis* DSM 11,300 was codon-optimized and synthesized by General Biosystems (USA). With the restriction sites *Nde*I and *Hind*III, the DgAS gene was synthesized with a 1984 bp size.

### 2.3. Protein Expression and Analysis

The seed culture of recombinant *E. coli* BL21 (DE3) harboring DgAS was prepared in LB broth medium supplemented with ampicillin antibiotic and incubated in a shaking incubator (200 rpm) at 37 °C overnight. Then, 500 μL of seed culture was added to a fresh 100 mL of LB medium with the 100 μL ampicillin antibiotic in a 500 mL conical flask and incubated at 37 °C for 24 h in a shaking incubator, without using IPTG for the induction. Cell pellets were then harvested by centrifugation at 842× *g* (3000 rpm) for 10 min, washed (vortexed, followed by centrifugation) with buffer (200 mM Tris-HCl and 10% glycerol of pH 7) twice, and re-suspended with 1 mL of the same buffer. Cell pellets were lysed using a Fisher Scientific Sonic Dismembrator Model 500 (5–9 s pulse on and off, total 360 s, at 20% amplitude) in an ice bath. The clear lysate was collected by high-speed centrifugation at 12,000 rpm (13,475× *g*) for 30 min at 4 °C. The crude protein obtained was

further analyzed by 12% (*w/v*) sodium dodecyl sulfate-polyacrylamide gel electrophoresis (SDS-PAGE). The soluble fraction of DgAS was purified using nickel nitrilotriacetic acid (Ni-NTA) affinity column chromatography (Qiagen Inc., Germantown, MD, USA) and eluted with different concentrations (10, 100, 200, and 500 mM) of imidazole, respectively, as reported previously [30]. Crude protein concentration was determined by using the Bradford method (Bradford et al. 1976), with bovine serum albumin as the standard.

### 2.4. Enzymatic Reaction

The glycosylation reaction was performed using resveratrol as an acceptor molecule in the presence of sucrose as a glucose donor catalyzed by amylosucrase (DgAS). A 200 μL of the total reaction mixture contained 0.2 mM resveratrol, 50 mM sucrose, 2 μg/mL amylosucrase enzyme, and 200 mM Tris-HCl buffer at pH 7. Resveratrol was dissolved in Dimethyl sulfoxide (DMSO), while sucrose was dissolved in water to create the required stock solution. The reactions were incubated at 40 °C. Reaction samples were obtained at 0.5 h, 1 h, 2 h, 3 h, and 4 h. Further reactions were terminated by treating the samples with chilled methanol, and the samples were then centrifuged at $13,475\times g$ for 30 min to remove the denatured proteins. After that, the samples were analyzed by high-performance liquid chromatography-photo diode array (HPLC-PDA) and confirmed by electrospray ionization mass spectrometry (ESI/MS) and nuclear magnetic resonance (NMR) analyses. Assay mixtures lacking enzymes and substrate resveratrol served as negative and positive controls, respectively.

### 2.5. Optimization In Vitro Reaction

Resveratrol concentration: Experimental conditions were modified by varying the resveratrol concentrations (0.5, 1.0, 1.5, 2.0, 2.5, 5.0, 10.0, and 15.0 mM), while keeping other components constant, as mentioned above. The concentrations of DgAS were maintained at 2 μg/mL in all reactions. The reactions were analyzed after incubation at 40 °C for 3 h. The reaction samples were analyzed by reverse-phase HPLC after 3 h by quenching with chilled methanol.

Sucrose concentration: Experimental conditions were modified by varying the sucrose concentrations (25, 50, 100, 150, 200, and 250 mM), while keeping other components constant, as mentioned above. The concentrations of DgAS were maintained at 2 μg/mL in each reaction. Resveratrol was used at a 0.5 mM concentration. The reactions were analyzed after 3 h to determine the effect of varying the concentrations of sucrose.

Enzyme concentration: Different identical sets of reaction conditions were set up by modifying the concentration of the DgAS. The concentrations of purified DgAS enzyme were maintained as 0.5, 1.0, 1.5, 2.0, 2.5, 3.0, 4.0, and 5.0 μg/mL in separate reaction sets. Reaction mixtures containing 0.5 mM resveratrol, 100 mM sucrose, and 200 mM Tris-HCl (pH 7) buffer at pH 7 were incubated at 40 °C for 3 h.

Temperature: Similarly, identical sets of reactions were carried out in 200 μL total volume using 200 mM Tris-HCl (pH7), 0.5 mM resveratrol, and 100 mM sucrose (final concentration) in the reaction mixture. During the reactions, the concentration of DgAS was maintained at 2 μg/mL. Reaction assays were incubated at 25, 30, 35, 40, 45, 50, 55, and 60 °C for 3 h.

Buffer or pH optimization: To determine the optimal pH, identical sets of reaction conditions were created, as in the temperature optimization reactions, while varying the pH of phosphate buffer in the range 5 and 6, using the Tris-HCl buffer with a pH 7 and 8, and the glycine buffer with a pH of 9. The reactions were carried out at 40 °C for 3 h.

### 2.6. Reaction with Different Sugar Donors and a Commercial Enzyme

Separately, identical sets of reactions were created using the same reaction conditions as those listed above. Instead of sucrose, other sugar donors such as starch, maltodextrin (4~7), maltodextrin (13~17), and maltodextrin (16.5~19.5) were used, along with cyclodextrin glucanotransferase (CGTase, from *Bacillus* sp.), a commercial enzyme.

### 2.7. In Vivo Preparation of Resveratrol Glucosides

A seed culture of recombinant *E. coli* BL21 (DE3) harboring DgAS was prepared in LB broth medium supplemented with ampicillin antibiotic and incubated in a shaking incubator (200 rpm) at 37 °C overnight. Then, 500 μL of seed culture was added to a fresh 100 mL of LB medium with the 100 μL of ampicillin antibiotic in a 500 mL conical flask. The culture was incubated at 37 °C for 24 h in a shaking incubator, without using IPTG for the induction. After that, 1 mM resveratrol and 100 mM sucrose feeding were done, and the mixture was incubated for 36 h at 37 °C in a shaking incubator at 200 rpm. During incubation, samples (1 mL each) were collected at 10 h and 36 h, and the reactions were quenched by adding 1 ml of chilled ethanol, followed by vortexing for 10 min. The aliquots were then centrifuged at 12,000 rpm (13,475× *g*) to remove the denatured protein. The samples were then analyzed by RP-HPLC.

### 2.8. Preparative Scale Production of Resveratrol Glucosides

The large-scale production reaction was carried out by using 1 mM resveratrol, 100 mM sucrose, 200 mM Tris-HCl (pH7), and 2 μg/mL DgAS in the reaction mixture. The total volume of the reaction mixture was 20 mL in a 50 mL tube. The reaction was incubated in a shaking incubator at 40 °C for 5 h. Samples were analyzed at different time points—3 h, 4 h, and 5 h. The reaction was terminated by boiling for 5 mins in a water bath to denature the enzymes. After that, the mixture was centrifuged at 12,000 rpm (13,475× *g*) for 15 min. The supernatant was concentrated using a rotatory evaporator and then used for the purification of products by preparative HPLC.

### 2.9. Analytical Methods

Reverse-phase HPLC-PDA analyses were performed at 280 nm UV absorbance using a $C_{18}$ column (Mightysil RP-18 GP (4.6 × 250 mm, 5 μm) (Kanto Corporation, Postland Oregon, USA)) connected to a photodiode array (PDA). HPLC was performed using a binary condition made up of water (0.1% trifluoroacetic acid (TFA)) and 100% acetonitrile (ACN), maintained at a flow rate of 1 mL/min for the 25 min program. The flow rate of ACN concentrations was set throughout as 0–40% (0–15 min), 40–75% (15–20 min), and 75–0% (20–25 min). The products were quantified using the authentic standard resveratrol substrate's curve created using different concentrations (0.625, 1.25, 2.5, 5, 10, 20, and 40 μM). The exact mass of the products was analyzed using high-resolution quadrupole time-of-flight electrospray ionization mass spectrometry (HR-QTOF-ESI/MS) [ACQUITY (UPLC, Waters, Milford, MA, USA)-SYNAPT G2-S (Waters)] in the positive ion mode. The compounds were purified using preparative-HPLC with a $C_{18}$ column (YMC-PACK ODS-AQ (250 × 20 mm I.D., 10 μm particle size) connected to a UV detector (280 nm) under a binary condition of $H_2O$ (0.05% TFA) and 100% ACN at a flow rate of 10 mL/min for 35 min. The ACN concentrations were 10, 30, 50, 90, and 10% for 0–15, 15–20, 20–25, 25–32, and 32–35 min, respectively. The purified product was concentrated using a rotatory evaporator, followed by lyophilization. The purified compounds were structurally determined by an Avance II 300 Bruker (Hardtstrabe, Karisruche, Germany) BioSpin NMR spectrometer equipped with a TCI CryoProbe (5 mm). All the samples were exchanged with $D_2O$ and dissolved in DMSO-$d_6$ for nuclear magnetic resonance (NMR) analysis. The compounds were further characterized by a 700 MHz Avance II 900 Bruker BioSpin NMR spectrometer (Hardtstrabe, Karisruche, Germany), using a Cryogenic TCi probe (5 mm). One-dimensional NMR ([1]H-NMR, [13]C-NMR) was performed to elucidate the structure of the compounds. All the raw data were processed using TopSpin 3.1 software (Bruker) and further analyzed using MestReNova 8.0 software (Mestrelab Research S. L., Feliciano Barrera, Bajo, Santiago de Compostela, Spain).

### 2.10. Assay of Resveratrol Glucosides for Cosmetic Activities

Cytotoxicity: Raw 264.7 cells, macrophage cell line, were seeded into 96-well culture plates at a density of $1.0 \times 10^4$ cells/well. After 24 h, the cells were treated with

resveratrol and resveratrol-*O*-glucoside, at indicated concentrations, in a medium containing 10.0% fetal bovine serum (FBS; Gibco, Thermo Fisher, Waltham, MA, USA) for 48 h. HS68 cells (human fibroblast cells) were seeded into 96-well culture plates at a density of $0.5 \times 10^4$ cells/well. After 24 h, the cells were treated with resveratrol and resveratrol-*O*-glucoside, at various concentrations (1.563–400 μM) in a medium containing 2.0% of FBS (Samchun, Pyeongtaek-si, Republic of Korea) for 48 h. B16 cells, a mouse melanoma cell line, were seeded into 96-well culture plates at a density of $8.0 \times 10^3$ cells/well and cultured overnight. The cells were then treated with resveratrol and resveratrol-*O*-glucoside, at various concentrations, in a medium containing 5.0% bovine serum (FBS; ATCC, Manassas, VA, USA) and a 1% penicillin-streptomycin mixture (PS' Lonza, Basel, Switzerland) for 72 h.

Cell viability was determined using a Quanti-MAX™ WST-8 cell viability assay kit (Biomax, Guri-si, Republic of Korea), according to the manufacturer's protocols. After treatment with 10.0% of WST-8 solution was added into each well from which the medium was removed. After incubation for 1 h, the plate was measured at 450 nm using a fluorescence microplate reader.

Anti-inflammation: Raw 264.7 cells, a macrophage cell line, were seeded into 96-well culture plates at a density of $1.0 \times 10^4$ cells/well. After 24 h, Raw 264.7 cells were treated with resveratrol and resveratrol-*O*-glucoside, at various concentrations (0.625, 1.25, 2.5 μM), in the presence of LPS (100 ng/mL) for 48 h in the medium, in the absence of phenol red containing 10.0% fetal bovine serum (FBS; Gibco, Thermo Fisher, USA). After incubation, the cell supernatants were collected from each well, and the NO contents were determined using a Griess reagent system kit (Promega, USA), per the manufacturer's protocol. The plate was measured at 540 nm using a fluorescence microplate reader. The results were normalized to the NO standard.

Anti-aging: HS68 cells (human fibroblast cells) were seeded into 24-well culture plates at a density of $2.0 \times 10^4$ cells/well. After 24 h, the cells were treated with resveratrol and resveratrol-*O*-glucoside, at various concentrations (3, 6, 12 μM), in a medium containing 2.0% FBS (Samchun, Republic of Korea) for 48 h. For comparison, transforming growth factor beta 1 (TGF-β1) (20 ng/mL) was used as a positive control. After incubation, the cell supernatant was collected from each well, and the pro-collagen content was determined using a human pro-collagen I alpha 1 ELISA assay kit (R&D Systems Inc., Minneapolis, MN, USA), following the manufacturer's protocol. The plate was measured at 450 nm using a fluorescence microplate reader. The results were normalized against the pro-collagen standard.

Whitening: B16 melanoma cells were seeded into 48-well cell culture plates at a density of $2.5 \times 10^4$ cells/well and incubated overnight. To induce melanin production, B16 cells were incubated with various concentrations of the test compounds in the presence of α-MSH (0.1 μM) in a phenol red-free cell culture medium in 48-well plates. After 72 h, the extracellular melanin content was measured at 405 nm using a Synergy™ HTX Multi-Mode Microplate Reader (Winooski, VT, USA) by transferring 100 μL of the medium in which the B16 cells were cultured to 96-well plates. Intracellular melanin content was calculated and corrected according to protein concentration. Control cells are considered to have 100% intercellular melanin content. To determine protein concentration, 200 μL of 1 N NaOH was added to the dissolve cells at 60 °C for 30 min. After that, 100 μL of cell lysate was added to a 96-well plate, and absorbance was measured at 450 nm using a microplate reader. The protein concentration of each sample was determined by a Pierce™ BCA Protein Assay.

Antioxidant: The DPPH((2,2-diphenyl-1-picrylhydrazyl) radical scavenging activity was measured to determine the antioxidant effect of each substance. After 100μL of 0.4 mM DPPH (Sigma Aldrich, USA) solution was added to 100 μL of a sample, the plate was incubated in a dark room for 15 min. The absorbance was then measured at 520 nm on a microplate reader. L-ascorbic acid (Samchun, Republic of Korea) was used as a positive control.

## 3. Results

### 3.1. In Vitro Reaction of Resveratrol

Protein expression and purification: To produce soluble recombinant proteins for in vitro reaction, pCHE-DgAS (Figure S1 of the Supplementary Information) was transformed into an *E. coli* BL21 (DE3) host. SDS-PAGE analysis of a soluble fraction of protein DgAS showed a clear band at around 72 kDa (Figure S2). The band size corresponds to the calculated molecular weight of a hexahistidine-tagged fusion protein. The soluble lysate-containing proteins were then subjected to purification using $Ni^{++}$-NTA beads. The purified proteins were concentrated, quantified, and used for in vitro reactions.

Glycosylation of resveratrol: The reaction mixture of resveratrol with sucrose and DgAS was first analyzed by RP-HPLC-PDA. Four different major peaks, P4, P3, P2, and P1, appeared in the chromatograms at a retention time ($t_R$) of 11.80, 12.03, 12.56, and 13.01 min, respectively (Figure 2A). The UV-VIS analysis of the peaks resembled those of resveratrol, with a subtle difference in the pattern (Figure 2B). The same sample was subjected to high-resolution QTOF-ESI/MS analysis. Further analysis showed that the exact mass $[M+H]^+$ $m/z^+$ of the product was 391.1381, which was a single glucose-conjugated resveratrol. The calculated mass of single sugar-conjugated resveratrol with the formula $C_{20}H_{23}O_8^+$ in the protonated form $[M+H]^+$ $m/z^+$ was 391.1387 Figure 2(Ci). Along with this, the di-glycosylated, tri-glycosylated, and tetra-glycosylated masses of resveratrol were observed as 553.1902, 715.2429, and 877.2952, respectively (Figure 2C). These masses exactly matched the calculated $[M+H]^+$ $m/z^+$ masses for the molecular formula of $C_{26}H_{33}O_{13}^+$, $C_{32}H_{43}O_{18}^+$, and $C_{38}H_{53}O_{23}^+$, with the masses of 553.1916, 715.2444, and 877.2972, respectively, which were the corresponding masses of di-glycosylated, tri-glycosylated, and tetra-glycosylated resveratrol (Figure 2C). The overall conversion of resveratrol to glucosides was about 97.0%.

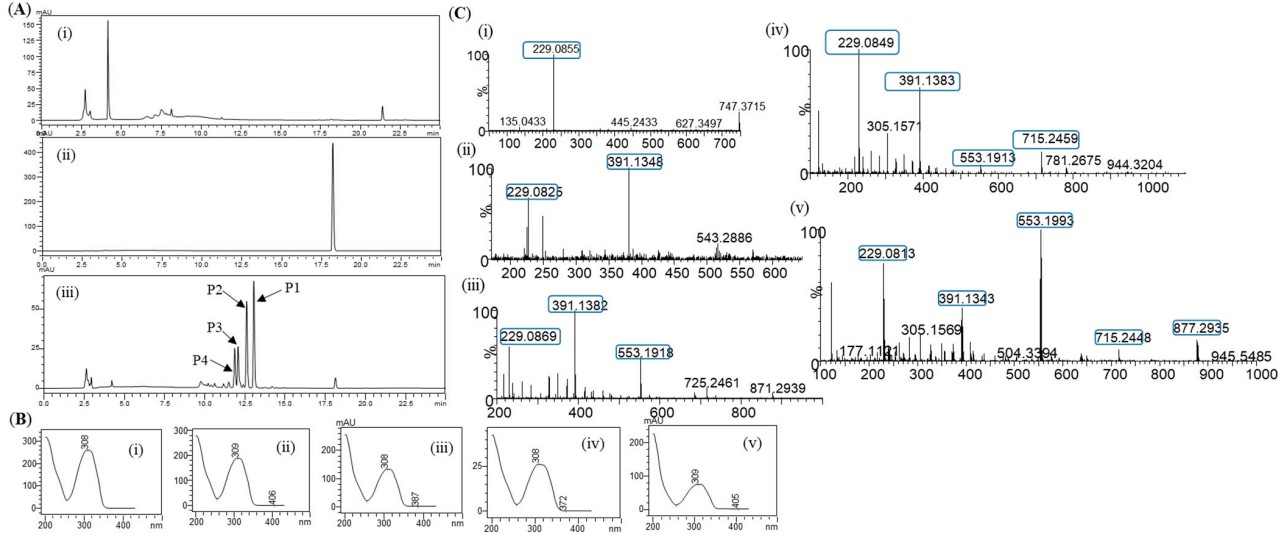

**Figure 2.** (**A**) HPLC analysis of resveratrol reaction mixture: (**i**) control reaction for regular glycosylation reaction of DgAS; (**ii**) standard resveratrol; and (**iii**) regular glycosylation reaction carried out using DgAS. (**B**) UV-VIS analysis of each major peak from HPLC: (**i**) standard resveratrol; (**ii**) P1; (**iii**) P2; (**iv**) P3; and (**v**) P4. (**C**) HRQTOF-ESI/MS analysis of (**i**) resveratrol standard; (**ii**) product P1; (**iii**) product P2; (**iv**) product P3; and (**v**) product P4.

During the reaction, samples taken at different time points were analyzed by RP-HPLC. The conversion of the resveratrol to its glucosides was increased up until the 3 h time point(~97.0%). It then remained constant, even though further incubation of the reaction (Figure S3). Therefore, a 3 h incubation time was taken as the optimized time for the in vitro reaction.

Resveratrol substrate tolerance: Eight sets of identical reactions were carried out at various resveratrol concentrations. The reactions contained 2 µg/mL DgAS, and the

other components of the reactions were fixed. Product formation and substrate conversion were monitored after 3 h. RP-HPLC analysis of the sample showed that when the concentration of resveratrol was 0.5 mM, maximum conversion of the substrate occurred (Figure S4). Therefore, 0.5 mM of resveratrol was taken as the optimized concentration for further reactions.

Optimization of sucrose concentration: Six sets of experiments were carried out using different concentrations of sucrose, while keeping all other reaction components constant. The results showed the maximum conversion of (~97.0%) resveratrol at 100 mM of sucrose. However, when the concentration of sucrose was 25 mM, the conversion was only about 80.0%. Moreover, with a concentration of sucrose more than 100 mM, there was a decrease in the conversion of the resveratrol (Figure 3i). These results showed an efficient supply of glucose for the glycosylation reaction. For further reactions, 100 mM of sucrose was used.

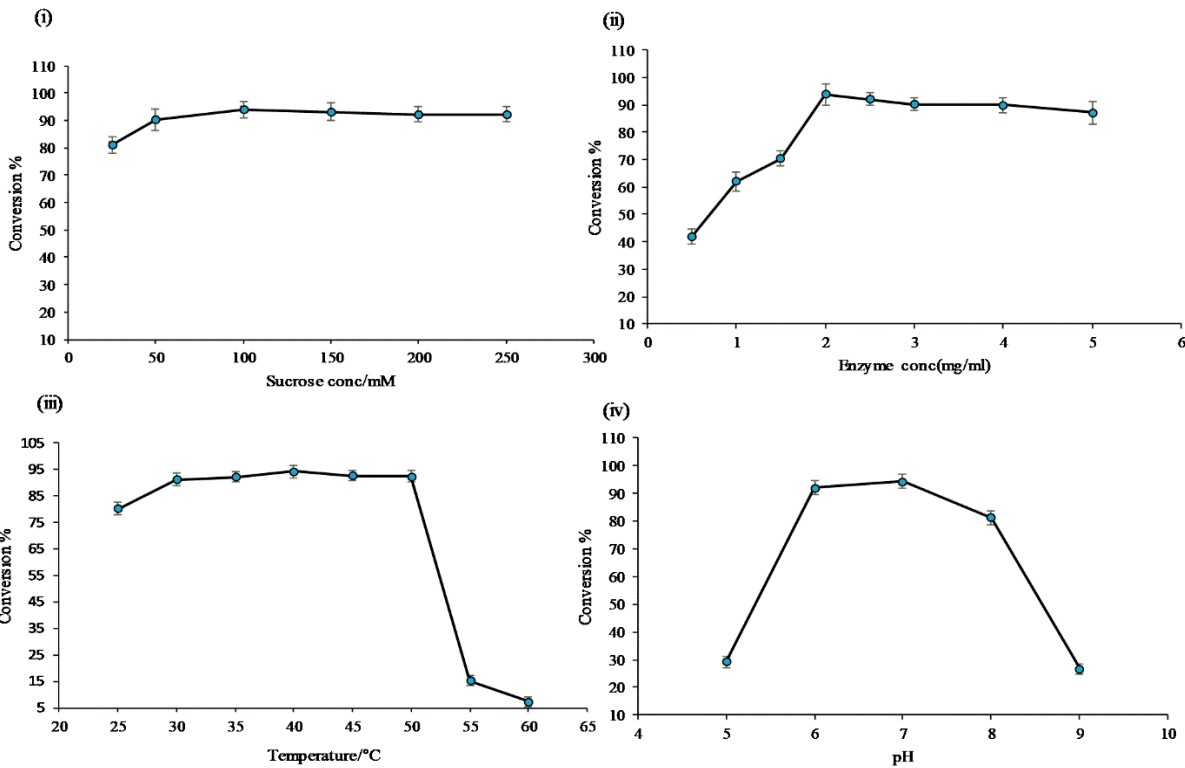

**Figure 3.** Conversion percentage of resveratrol to resveratrol glucosides in the optimization cascade reaction; (**i**) variation of sucrose concentration; (**ii**) different enzymes concentration of DgAS; (**iii**) variation of reaction temperature; and (**iv**) variation of buffer and pH.

Enzyme concentration: To identify the optimized concentration of the enzyme for the reactions, eight sets of identical reactions were carried out, and the outcome of the reactions were closely monitored to obtain the optimal conversion of resveratrol. As expected, a different pattern of the product profile was obtained when varying the enzyme concentration in the reaction. The conversion percentage of resveratrol at the concentration of 0.5 µg/mL was about 42.0%, which increased at a higher concentration of DgAS and reached about 90.0% at 2 µg/mL, decreasing slightly at 2.5 µg/mL, but slightly increasing at 3 µg/mL (Figure 3ii). Therefore, 2 µg/mL of DgAS was used for further reactions.

Effects of temperature: The results obtained at different temperatures, ranging from 25 to 60 °C, showed multiple impacts on product formation, as seen in Figure 3iii. At 25 °C, only two products, P1 and P2, formed, while at 55 and 60 °C, only one product, P1, formed. The maximum conversion of the resveratrol at 40 °C was found to be about 97.0%. All products (P1, P2, P3, and P4) formed at a temperature range of 30 to 50 °C. The optimal reaction temperature was 40 °C.

Effects of buffer: To find out the optimal pH for the reaction assay, glucoside production was investigated at 40 °C, keeping all other reaction ingredients intact, while changing the buffer and pH. The pH of the reaction was adjusted to 5 to 9 in 200 mM of phosphate, Tris-HCl, and glycine buffer, with their respective pH as mentioned in the Materials and Methods section. The reaction was monitored after incubating at 40 °C for 3 h. The results showed that at low pH levels of 5 and 9, there was less conversion of resveratrol (~20.0%) to its glucosides, as shown in Figure 3iv. The conversion of the resveratrol, which was about 97.0%, was the highest at pH 6 and 7. Therefore, the optimal pH for the reaction was 7. All products formed at pH 7 were found to be measurable. They were dominant under the reaction conditions (Figure 3iv).

### 3.2. Effect of using Different Sugar Donors and a Commercial Enzyme

After analyzing the results obtained from the RP-HPLC chromatograms of samples incubated at 40 °C for 3 h, using different sources of sugar donors and commercial enzyme CGTase, the maximum conversion of resveratrol was found using sucrose and DgAS (Figure S5). The conversion rate with DgAS and sucrose was found to be about 97.0%. With CGTase and different sugar donors (starch and different maltodextrins), the conversion rates were only around 20.0%.

### 3.3. Preparative Scale Production of Resveratrol Glucosides

When product profiles at different time points were analyzed, four distinct peaks were initially observed. Among them, the peak at $t_R$ 13.01 min (P1) was initially the most dominant. However, with increasing the incubation time from 3 h, to 4 h, to 5 h, peaks at $t_R$ 12.56 min (P2), 12.03 min (P3), and 11.80 min (P4) gradually increased, while that of P1 gradually decreased. The conversion of resveratrol to its glucosides was found to increase more significantly with an increasing incubation period in the large-scale reactions (Figure 4 and Figure S6). Based on the nature of product formation, the amount of the desired products can be controlled by changing the incubation period of the reaction. As time increased, the mono-glycosylated products changed into di-glycosylated or poly-glycosylated products. The overall conversion of the resveratrol to its glucosides was about 97.0%. The calculated conversion rates for regular products of P1, P2, P3, and P4 at 5 h were 38.8, 28.8, 16.2, and 12.3%, respectively (Figure 4).

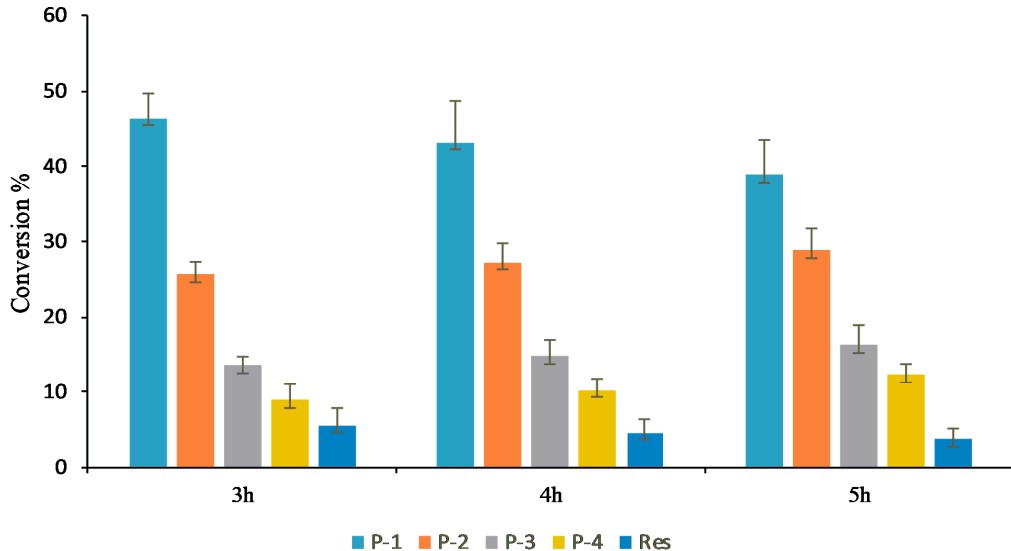

**Figure 4.** Conversion of resveratrol to its glucosides in the preparative scale in vitro reaction at different times points.

### 3.4. In Vivo Production of Resveratrol Glucoside

During the in vivo production of resveratrol glucosides, the first sample was removed at 10 h and analyzed by RP-HPLC. The chromatogram of the sample showed a single peak at a retention time ($t_R$) of 13.01 min (Figure 5(Aii)), which was exactly the same as the peak obtained in the in vitro reaction. Another sample was taken at 35 h and analyzed by RP-HPLC, as described earlier. It was found that the standard substrate was decreased, with more products peaking at retention times of 11.80, 12.03, and 12.56 min, as seen in Figure 5(Aiv).

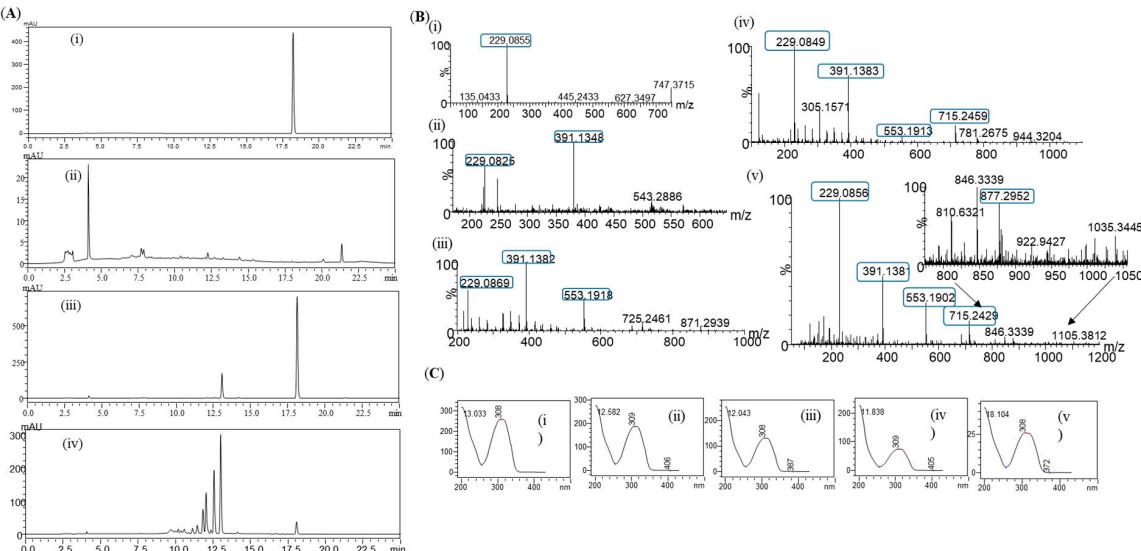

**Figure 5.** (**A**) HPLC-PDA chromatogram for resveratrol and resveratrol glucosides in the in vivo system; (**i**) standard resveratrol; (**ii**) control; (**iii**) the 10 h sample; and (**iv**) the 35 h sample. (**B**) HR-QTOF ESI/MS analysis of (**i**) resveratrol standard and (**ii**) resveratrol glucoside at 10 h, P1; (**iii**) P2; (**iv**) P3; and (**v**) P4. (**C**) UV-VIS analysis of (**i**) standard resveratrol; (**ii**) product, P1; (**iii**) P2; (**iv**) P3; and (**v**) P4.

When the same samples were subjected to high-resolution QTOF-ESI/MS analysis, the exact mass $[M+H]^+$ $m/z^+$ of a product was 391.1348, which resembled the single glucose conjugated resveratrol of the 10 h sample. The calculated mass of single sugar-conjugated resveratrol, with a formula $C_{20}H_{23}O_8^+$ in a protonated form $[M+H]^+$ $m/z^+$, was 391.1387 (Figure 5B,C). Moreover, with the 35 h sample, exactly the same pattern of the masses was obtained, with the peaks as seen in the in vitro sample analysis (Figure 2C). These results clearly indicate that resveratrol glucosides obtained by DgAS could also be formed in in vivo systems.

### 3.5. Structural Elucidation of Resveratrol Glycoside Products

Four major peaks produced during the preparative scale were purified using prep-HPLC. According to HPLC analysis, the purity level of each compound was greater than 95.0%. The purified compounds were dried using a rotatory vacuum evaporator, followed by lyophilization for preparing NMR samples as described in the Materials and Methods section. The HR-QTOF-ESI/MS peaks P1, P2, P3, and P4 showed the presence of a single, di-, tri-, and tetra-glycosylated mass in resveratrol, respectively (Figures 2C and 5B).

Furthermore, the $^1$H-NMR analysis of P1 showed the absence of one proton peak at a chemical shift value δ 8.18 ppm (3-OH), indicating that a glucose molecule was attached at the 3-OH position of resveratrol. The presence of an anomeric proton peak at δ 5.34 ppm (d, J = 3.7Hz) also confirmed the glucose unit at the alpha configuration. Similarly, in the $^{13}$C-NMR analysis, an anomeric carbon peak appeared at δ 97.77 ppm. Details of the $^1$H- and $^{13}$C- data are shown in Tables S1 and S2.

In the same way, in the $^1$H-NMR analysis of P2, two proton peaks showed chemical shift values of δ 8.18 ppm (3-OH) and δ 6.14 ppm (4'-OH), indicating that glucose was attached at the 3-OH and 4'-OH positions, respectively. In the $^{13}$C-NMR, an anomeric proton at δ 5.34 ppm (d, J = 3.7 Hz), an anomeric carbon peak at δ 97.76 ppm, and the protonated mass of two glucoses confirmed the structure of resveratrol as one-one glucose attached at the 3-OH and 4'-OH position of the resveratrol. However, after analyzing P3 and P4, the proton peaks at δ 8.18 ppm (3-OH) and δ 6.94 ppm (4'-OH), respectively, were missing. Further analysis of the peaks of both products at the similar anomeric proton position at δ 4.85 (d, J = 3.6 Hz), along with their masses, clearly indicated that P3 contained three glucose polymer chains at 4'-OH, and P4 exhibited four glucose polymer chains at the 3-OH position, with alpha configuration. The presence of anomeric carbon at δ 98.45 ppm further supported the structure of P3 and P4. All details of the $^1$H-NMR and $^{13}$C-NMR analyses are listed in Tables S1 and S2.

According to the 1D NMR and HR-QTF-ESI/MS analyses, the four compounds were identified as resveratrol-3-*O*-α-glucoside (P1), resveratrol-3,4'-*O*-α-diglucoside (P2), resveratrol-3-*O*-α-triglucoside (P3), and resveratrol-4'-*O*-α-tetraglucoside (P4) (Figure S7).

*3.6. Cosmetic Activities of Resveratrol Glucosides*

The effects of resveratrol-*O*-glucosides and resveratrol on cell viability were determined using Raw 264.7 cells, HS68 cells, and B16 cells, as mentioned above in the Materials and Methods section. After incubation for 72 h, the samples were tested. The results revealed that resveratrol-*O*-glucosides were less toxic to B16 cells than was resveratrol. After the cells were treated with 100 μM resveratrol-*O*-glucosides or 100 μM resveratrol, cell viability was around 50.0% or 30.0%, respectively (Figure 6A). When Raw 264.7 cells were treated with 200 μM resveratrol-*O*-glucosides or resveratrol for 48 h, cell viability was greater than 95.0%, or only around 10.0%, respectively (Figure 6B). Similarly, when HS68 cells were treated with 400 μM resveratrol-*O*-glucoside or the original resveratrol, cell viability was around 100% (Figure 6C). Therefore, the cytotoxicity of resveratrol glucosides is much lower than that of resveratrol.

Raw 264.7 cells (a macrophage cell line) were seeded into 96-well culture plates at a density of $1.0 \times 10^3$ cells/well, as mentioned in the Materials and Methods section. After incubation for 48 h, the samples were tested according to standard protocols, and the results were analyzed. The results clearly showed that both resveratrol and resveratrol-*O*-glucosides exhibited anti-inflammatory effects, with different IC$_{50}$ values (Figure 7A).

As mentioned above, HS68 human fibroblast cells were seeded in a 24-well culture plate at a density of $20 \times 10^4$ cells/well and incubated for 48 h. Supernatant samples were then collected to determine pro-collagen contents. The results showed that both resveratrol and resveratrol-*O*-glucosides exhibited anti-aging effects (Figure 7B).

As described in the Materials and Methods section, B16 melanoma cells were seeded in a 48-well culture plate at a density of $2.5 \times 10^4$ cells/well and cultured overnight. They were then incubated with control and resveratrol compounds for 72 h. These samples were then analyzed, according to the standard protocol mentioned above. It was found that both resveratrol and resveratrol-*O*-glucosides showed whitening effects, although they had different IC$_{50}$ values (Figure 8A).

The DPPH radical scavenging assay was carried out as described in the Materials and Methods section to determine the anti-radical power of the antioxidants. The samples were analyzed after incubation for 15 min at a 515 nm wavelength in a microplate spectrophotometer. It was found that both resveratrol and resveratrol-*O*-glucosides exhibited similar antioxidant activities, with different IC$_{50}$ values (Figure 8B).

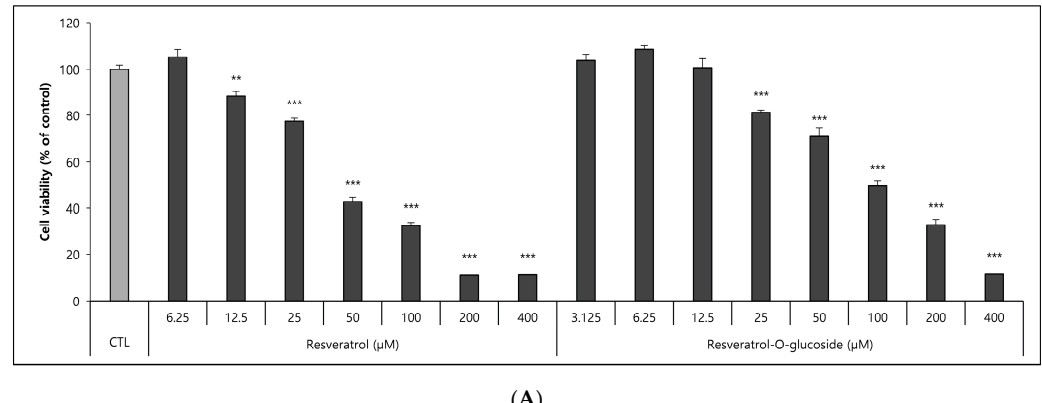

(**A**)

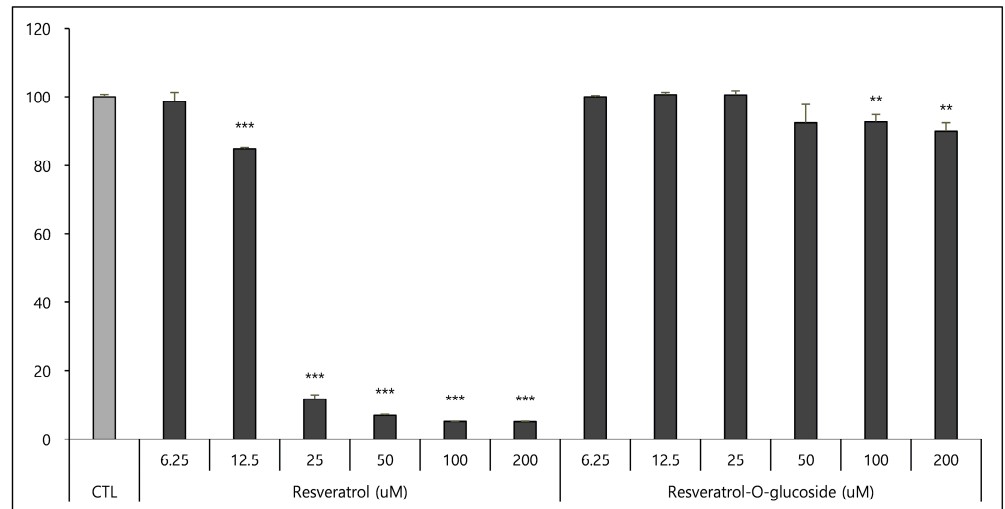

(**B**)

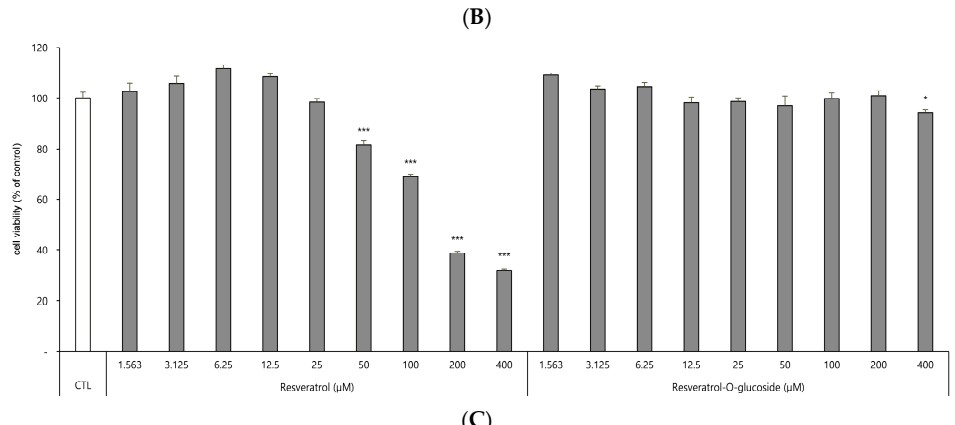

(**C**)

**Figure 6.** Cytotoxicity evaluation of raw resveratrol and resveratrol-*O*-glucoside. (**A**) Experimental conditions: B16/8 $\times$ 10$^3$ cells/96-well plate, 24 h incubation period after sample treatment. (**B**) Experimental conditions: Raw 264.7/1.0 $\times$ 10$^4$ cells/96-well plate, 48 h incubation period after sample treatment. (**C**) Experimental conditions: HS68 cells/0.5 $\times$ 10$^4$ cells/96-well plate, 48 h incubation period after sample treatment. Note—media containing 5% ATCC FBS was used for cell culturing; significance is indicated only when there is toxicity relative to CTL ***; $p < 0.001$/**; $p < 0.01$/*; $p < 0.1$.

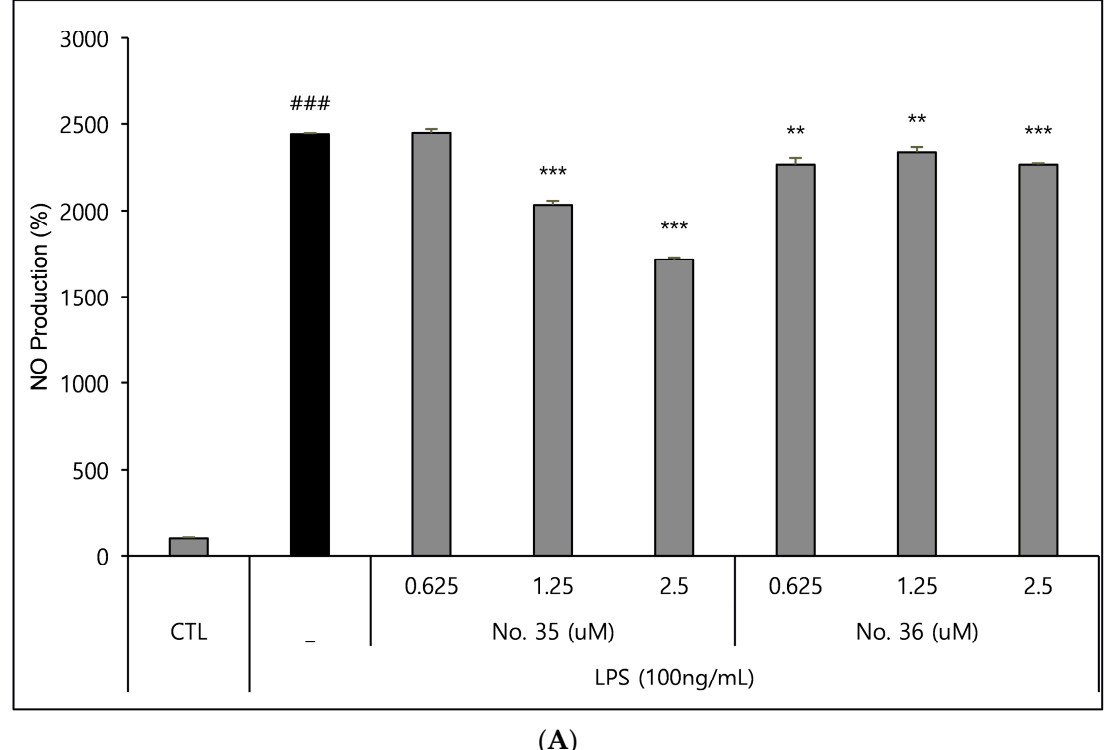

**(A)**

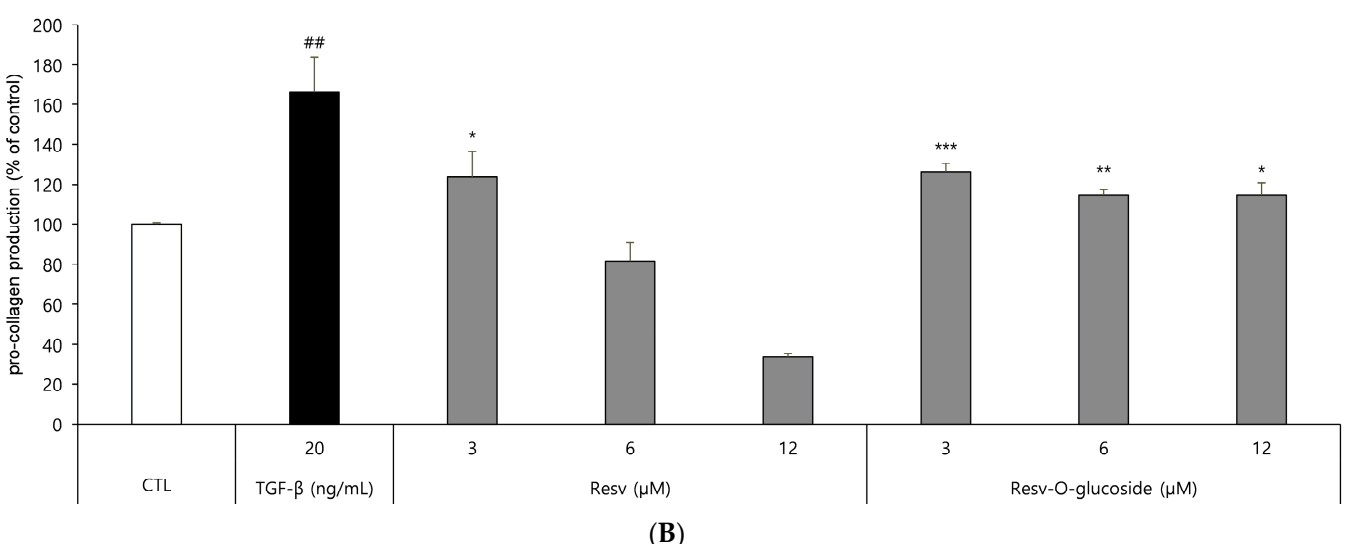

**(B)**

**Figure 7.** Anti-inflammation and anti-aging tests for resveratrol and resveratrol-*O*-glucosides. (**A**) Experimental condition: Raw 264.7/1.0 × $10^3$ cells/96-well plate, 48 h incubation period after sample treatment. (**B**) Experimental condition: HS68/3.0 × $10^4$ cells/24-well plate, 48 h incubation period after sample treatment. Note: For cell culture, triple FBS (10%) was used; for drug treatment, triple FBS (2%) was used. Significance is indicated only when there is efficacy over CTL. *** or ###; $p < 0.001$/** or ##; $p < 0.01$/*; $p < 0.5$. TGF-β is used as a positive control for procollagen production.

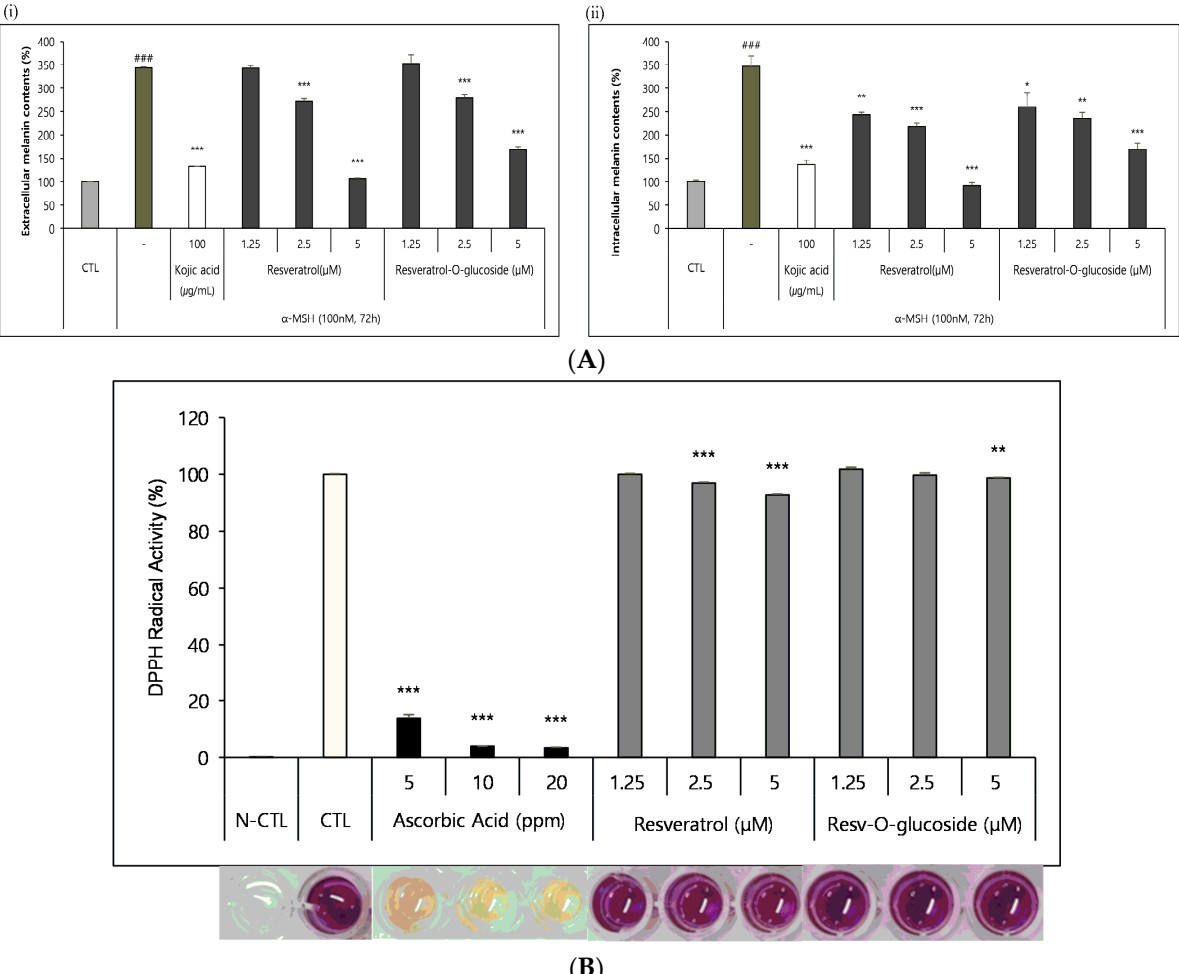

**Figure 8.** Whitening and antioxidant tests of raw resveratrol and resveratrol-*O*-glucoside. (**A**) (**i**) extracellular melanin contents (%), and (**ii**) intracellular melanin contents (%). Experimental conditions: B16/25 × 10³ cells/48-well plate/72 h incubation period after sample treatment. Note: media containing 5% ATCC FBS was used for cell culture. The stimulus is α-MSH, and the positive control is Kojic acid (a functional whitening ingredient). The inhibition of melanogenesis was calculated as CTL (100%, α-MSH alone (0%)). Significance is indicated only when there is a whitening effect. *** or ###; $p < 0.001$/**; $p < 0.01$/*; $p < 0.1$. (**B**) Experimental conditions: DPPH test B16/8 × 10³ cells/96-well plate, 15 min incubation after sample treatment. Note—The experimental concentrations were specified considering non-toxic concentrations in both Raw 262.47, HS68, and B16 cells. Significance is shown only when there is an antioxidant advantage over CTL ***; $p < 0.001$/**; $p < 0.01$.

## 4. Discussion

Resveratrol and its derivatives have attracted considerable interest and demand due to their potent biological activities in the medical and nutraceutical fields [3,5]. Recent advancements in biological tools and techniques has enabled the cost-effective and rapid production of such bioactive plant products in microbial hosts. Although microbial hosts lack resveratrol biosynthetic genes, they offer a promising alternative source for the large-scale production of resveratrol and its derivatives using metabolic engineering and synthetic biology approaches [37,38]. However, despite the use of the latest techniques, the overall production of resveratrol and its derivatives remains insufficient [39,40]. Therefore, optimizing each gene involved in the biosynthetic pathways is necessary to improve enzyme activity in trans-located hosts by supplying sufficient precursors and regulating resveratrol concentrations within the cell, which could enhance production [38,41]. Furthermore,

research has demonstrated that resveratrol derivatives exhibit higher antioxidant activities than does resveratrol [42]. Due to their higher stability compared to resveratrol in response to light, oxygen, and extreme pH, the production of resveratrol derivatives is increasing and becoming more desired, not only for their high medicinal, nutritional, and cosmetic values, but also for their superior stability properties.

In this research, we devised in vitro and in vivo biocatalytic systems using the enzyme amylosucrase (DgAS) derived from *Deinococcus geothermalis* DSM 11,300 for the efficient transglycosylation of resveratrol, with sucrose as the glycosyl donor [3,30]. Amylosucrase is a versatile enzyme with several industrial applications such as hydrolysis, polymerization, isomerization, and transglycosylation [43]. During the process of sucrose hydrolysis, the $\alpha$-1, $\beta$-2-glycosidic bond of sucrose breaks down, leading to the formation of the sucrose isomers of turanose and trehalulose. Similarly, in isomerase activity, sucrose isomerizes to produce turanose and trehalulose, while the transglycosylation activity produces $\alpha$-1,4-glucans, such as amylose polymers [44,45]. DgAS can transfer the glucosyl moiety from sucrose to glucose or fructose, without using costly UDP-glucose as a substrate [30]. However, the transglycosylation of resveratrol mostly occurs at the C-3 or C-4' position, rather than at the C-5 position, which is sterically hindered, due to position-specific transglycosylation, [46]. Studies have shown that DgAS attaches four or fewer glucose units to polyphenols during transglycosylation [46]. Our results indicated the formation of four major products, with product P4 exhibiting a four-glucose-unit chain, attached to the 4'-OH position of resveratrol; product P3, displaying a three-glucose-polymer chain at the 3-OH position; product P2, containing glucose moieties at both the 3-OH and 4'-OH positions, with a di-glycosylated mass; and product P1, identified as resveratrol-3-*O*-$\alpha$-glucoside (Figure S7). Although DgAS exhibited high specific activity and thermostability, its polymerization activity was relatively lower than those of other ASases, owing to its exceptional thermostable characteristics with a half-life of 6 h at 55 °C [47,48]. The melting temperature of DgAS was 61.4 °C higher, compared to other microbial amylosucrases (ASs) [35]. Because of many a heavy network of hydrogen bonds and strong salt bridge interactions, DgAS form dimer structures that result in higher thermal stability. Moreover, when DgAS was utilized to transglycosylate arbutin, it transferred a glucose unit of sucrose to the C-4 position in the glucose residue of salicin and arbutin, with the major arbutin glycoside transfer product being identified as glucosyl salicin [49,50].

Although both in vitro and in vivo systems are effective in producing resveratrol-*O*-glucosides, each has its advantages and disadvantages. In vitro systems using DgAS are fast, direct, easy to handle, environmentally friendly, and require fewer steps for product extraction and purification [30,37]. However, these systems require large amounts of protein and a longer time for the purification of enzymes for scale-up production. In contrast, in vivo systems do not require additional protein purification. However, they require a longer time for culture before product harvest. They may also exhibit additional bacterial product peaks that can create an extra burden for purification. Ultimately, the choice between in vitro and in vivo systems for producing bioactive resveratrol-*O*-glucoside or resveratrol poly-glucosides depends on the available facilities and the specific production requirements. Moreover, by immobilizing DgAS with Amicogen LKZ118 beads, the DgAS was reused 50 times. Among these cycles of reuse, at up to 35 reuse cycles, the efficiency of the enzyme remained greater than 90%, decreasing gradually. Proper covalent binding with Amicogen beads and an adequate number of functional groups of DgAS enhance the stability and efficiency of the process by retaining the active conformation of the enzyme. However, reaction under aqueous and radical reaction conditions leads to the loss of activity and durability of the enzyme [30,33].

Aging is a natural biological process that poses a threat to living organisms by making them susceptible to various diseases and disorders. The skin is the largest and most exposed organ of the body. It is highly vulnerable to aging, resulting in cancer, skin damage, and other harmful conditions [51]. Resveratrol and its related compounds have been found to exhibit significant protection against skin aging by inhibiting tyrosinase, an enzyme

involved in melanin production. Melanin regulates vitamin D3 biosynthesis and enhances the skin's resistance to tumors and sunburns [52,53]. Therefore, tyrosinase inhibitors, such as resveratrol and its glucoside derivatives, are highly sought-after in the medicinal and cosmetic industries as depigmentation agents [54].

During melanin biosynthesis, the intervention of L-cysteine can lead to the formation of pheomelanin, which can be cytotoxic, as it produces reactive oxygen species (ROS) [55,56]. Hence, controlling excess melanin and intercellular ROS is a crucial biochemical step in developing a safe and effective skin-whitening agent [57]. More than ten skin-whitening substances, such as ascorbic acid, kojic acid, and arbutin, have been used as cosmetic ingredients [57]. Reports have suggested that resveratrol and its glycosylated derivatives possess antioxidant and free radical scavenging capabilities, as they can promote the activities of various antioxidative enzymes [58]. The presence of phenolic hydroxyl groups and electron delocalization across the molecules are the primary mechanisms of their antioxidant properties [59].

The secretion of a biological response modifier in response to external or internal stimuli can result in allergies and inflammation in the body. Macrophages are the primary source of cytokines and growth factors that can affect epithelial, endothelial, and mesenchymal cells, leading to inflammation [60]. Under normal physiological conditions, a moderate increase in inducible NO synthase (NOS) activity leads to NO formation, which has bactericidal effects. However, high levels of NO and its derivatives, such as peroxynitrite ($ONOO^-$), can cause inflammation. Resveratrol preincubation with Raw 264.7 cells decreased the presence of inflammation by downregulating iNOS mRNA and protein [61,62]. Due to its high thermostability and specific activity among all known ASases, *D. geothermalis* ASase (DgAS) is highly significant in the industry through its transglucosylation activity among different acceptor substances [45]. Such variations in transglucosylation activity might be due to changes in the active site conformation caused by intramolecular interactions, including hydrogen bonds, disulfide bonds, and ionic interactions [45,50]. Although many flavone materials are available as potential pharmaceutical candidates, their application is restricted because of their poor solubility. Thus, the main aim of this study was to synthesize flavone glycosides on an industrial scale using DgAS [63,64].

## 5. Conclusions

In summary, the enzymatic production of resveratrol glucosides using DgAS, in both in vitro and in vivo systems, is highly effective, with a conversion rate of approximately 97.0% in a short period of time. DgAS can add glucose to multiple hydroxyl positions and conjugate glucose in the chain, resulting in highly soluble glycosylated products. Furthermore, positive outcomes obtained from in vitro assays using Raw 264.7, HS68, and B16 cell lines highlight the potential application of resveratrol glucosides in the cosmetics industry. Therefore, this system holds promise for the cost-effective industrial production of resveratrol glucosides.

**Supplementary Materials:** The following supporting information can be downloaded at: https://www.mdpi.com/article/10.3390/cosmetics10040098/s1, Figure S1: Vector map of recombinant construct used in this study. Figure S2. SDS-PAGE of protein DgAS. M: marker; Lane 1: Control BL-21(DE3); Lane 2: crude enzyme produced in E. coli BL-21(DE3); Lane 3: purified enzyme. Figure S3. Conversion percentage of resveratrol glucosides at different time interval. Figure S4. HPLC analysis of resveratrol reaction mixtures in different concentrations of resveratrol. Figure S5. Conversion of resveratrol to its glucosides with different sugar donors and commercial tansglycosyaltion enzyme, CGTase. Figure S6. HPLC analysis of large scale in vitro resveratrol reaction mixture at different time points;- (i) 3 h, (ii) 4 h, (iii) 5 h, and (iv) Standard resveratrol. Figure S7. Structural Elucidation of resveratrol-*O*-glucosides products. Table S1. 1H-NMR analysis of resveratrol glucosides. Table S2. 13C-NMR analysis of resveratrol glucosides.

**Author Contributions:** Conceptualization, J.K.S.; Methodology, S.B.T.; Validation, S.B.T.; Formal analysis, B.G.P.; Investigation, J.J., B.G.P. and D.S.; Resources, J.J.; Data curation, S.B.T., J.J., D.S. and C.S.L.; Writing—original draft, S.B.T.; Writing—review & editing, J.K.S.; Supervision, J.K.S.; Funding acquisition, J.K.S. All authors have read and agreed to the published version of the manuscript.

**Funding:** This work was supported by the National Research Foundation of Korea (NRF) grant funded by the Korean government (MEST) (NRF-2021R1A2C2004775).

**Institutional Review Board Statement:** Not applicable.

**Informed Consent Statement:** Not applicable.

**Data Availability Statement:** Not applicable.

**Conflicts of Interest:** The authors declare no conflict of interest.

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
