# Peer review of "Production of Resveratrol Glucosides and Its Cosmetic Activities"

_cosmetics, doi:10.3390/cosmetics10040098_

Round 1

Reviewer 1 Report

the paper describe the application the enzymatic production of resveratrol glucosides using DgAS in both in 556 vitro and in vivo systems with a very good yields and best solubility of the final product. and this is is original especially since the final product also showed good cosmetic activity.

No coments

Author Response

We are grateful to the reviewer for thoroughly reviewing our manuscript and for his kind words.

Reviewer 2 Report

The current work focuses on the Production of resveratrol glucosides and its cosmetic activities. The experimental work appears to have been carried out well. However, a few points deserve attention for further publication. I suggest that it is accepted for publication after the following revisions:

- ABSTRACT: Production of resveratrol glucosides and its cosmetic activities: What parameters were optimized? Authors must include numbers with the results found. Stability of amylosucrase DgAS? How much amylosucrase DgAS were utilized to process? Furthermore, what are the conditions of reactions? Temperature, pH, ionic strength, for example. This information should be included in the abstract.

- INTRODUCTION:

- In this study, Production of resveratrol glucosides and its cosmetic activities: physical and covalent were implied for preparation? What the advantages? Additionally, the spacer arm, the steric hindrances of amylosucrase DgAS reaction caused by this groups when compared to the others groups? These strategies used should be better explained in the manuscript.

- Amylosucrase DgAS are very special, having a peculiar mechanism of action. This information must be clear in the introduction to present manuscript.

- A paragraph describing the properties, application, mechanism of actuation to Amylosucrase DgAS must be included in the manuscript.

- What is the origin of this Amylosucrase DgAS? Is it a commercial ? Modified? How was it produced?

- Amylosucrase DgAS: Improved kinetic, and efficient increased the activity? This process needs to be explained in the introduction of the manuscript.

- Amylosucrase DgAS: What optimization strategy was used? Why was it used? This information needs to be explained in the introduction of the manuscript.

- Amylosucrase DgAS presented were compared with a commercial material? This information must be clear in the introduction.

- The contribution and importance of these studies in the work performed must be explained in the introduction of the manuscript.

MATERIALS:

- Include the concentration of solutions.

METHODS:

- Include the molar concentration of all the chemicals used, the way the methods are presented, not possible reproducibility.

- Amylosucrase DgAS: Please include more details, temperature, pH, molar ratio, ionic strength.

- RESULTS AND DISCUSSION:

- The influence of substrate systems to Amylosucrase DgAS stability was also investigated? The Amylosucrase DgAS showed how about stability?

- The thermal stability to Amylosucrase DgAS prepared is one of the most important application criteria for diferent applications. This stability depends to Amylosucrase DgAS preparation strategy. It also depends on the stabilization of the Amylosucrase DgAS. This discussion could be improved. Please include in the manuscript.

- The stability in organic solvents, metal ions, or detergent enables its wide application in synthesis processes which nowadays are in great demand from the point of view of industrial. The effect of organic solvents on the Amylosucrase DgAS activity was studied? For example, in the presence of organic solvents?

- Was determined the full loading of Amylosucrase DgAS prepared under the optimized conditions? This information must be clear in the manuscript.

- The Amylosucrase DgAS may experience protein aggregation (mainly near to the isoelectric point). This may be caused by undesired Amylosucrase DgAS - interactions where inactivation that can stabilize incorrect Amylosucrase DgAS structures.

- The optimization of Amylosucrase DgAS preparation process, the preparations shown having diffusion limitations? Considering the strategy presented in this manuscript. Please, this should be explained in the manuscript. What were the optimum conditions?

- Effect of solution pH since the solution pH affects the generation of hydroxyl radicals and also influences the surfasse charge and interface potential properties of the catalyst, it is one of the important factors. Amylosucrase DgAS showed considerable improves in the kinetic parameters in terms of activity, specific activity, Km and Vmax, optimum pH and Temperature?

- Reusability of Amylosucrase DgAS:  The reusability of Amylosucrase DgAS particles is essential while considering reactions. Amylosucrase DgAS reusability was accounted for? Reusability studies showed that the remaining Amylosucrase DgAS assay was obtained to reduce with the increasing number of re-use cycles. The reusability of Amylosucrase DgAS without alteration in its load capacity of performance with the resulting is an advantage. After cycles, please, explain these results. What other factors can influence the results achieved? In addition, the results should be compared with other works of literature in the same application line.

- Enhanced stability of Amylosucrase DgAS: Other factors that cause the loss of durability and stability should be explained in the manuscript.

- Please, check all references according to the author's instructions.

- Include more details in the figures (error bars) and tables captions.

- The manuscript must be formatted according to the journal's standards.

Minor editing of English language required

Author Response

The current work focuses on the Production of resveratrol glucosides and its cosmetic activities. The experimental work appears to have been carried out well. However, a few points deserve attention for further publication. I suggest that it is accepted for publication after the following revisions:

 - ABSTRACT: Production of resveratrol glucosides and its cosmetic activities: What parameters were optimized? Authors must include numbers with the results found. Stability of amylosucrase DgAS? How much amylosucrase DgAS were utilized to process? Furthermore, what are the conditions of reactions? Temperature, pH, ionic strength, for example. This information should be included in the abstract.

Response;- Thank you so much for the suggestions. However, we try to make our abstract short and effective as far as possible. But, we have mentioned all these here.

- INTRODUCTION:

- In this study, Production of resveratrol glucosides and its cosmetic activities: physical and covalent were implied for preparation? What the advantages? Additionally, the spacer arm, the steric hindrances of amylosucrase DgAS reaction caused by this groups when compared to the others groups? These strategies used should be better explained in the manuscript.

Response;- We appreciate the suggestions. So, we have added information on the immobilization of enzymes in the manuscript as stated in red texts.

 - Amylosucrase DgAS are very special, having a peculiar mechanism of action. This information must be clear in the introduction to present manuscript.

Response;- As per the suggestion, We have briefly described the mechanism of the enzyme in our manuscript in the red texts.

- A paragraph describing the properties, application, mechanism of actuation to Amylosucrase DgAS must be included in the manuscript.

 Response;- We address the suggestions in our manuscript

- What is the origin of this Amylosucrase DgAS? Is it a commercial ? Modified? How was it produced?

Response;- About the origin of DgAS, has been mentioned in the abstract section as Deinococcus geothemalis DSM11300 and in the materials and methods section. It is not a commercial enzyme. It was produced in E.coli by heterologous expression.

- Amylosucrase DgAS: Improved kinetic, and efficient increased the activity? This process needs to be explained in the introduction of the manuscript.

Response;- It’s a good suggestion but we did not perform kinetic studies of the experiments. Therefore, in the future, we will include all these suggestions in similar types of work.

- Amylosucrase DgAS: What optimization strategy was used? Why was it used? This information needs to be explained in the introduction of the manuscript.

Response;- We have included it in the result section.

- Amylosucrase DgAS presented were compared with a commercial material? This information must be clear in the introduction.

 Response;- We have mentioned briefly this information in our result section.

- The contribution and importance of these studies in the work performed must be explained in the introduction of the manuscript.

 Response;- Thank you so much for the suggestions.

MATERIALS:

- Include the concentration of solutions.

Response;- The concentration of a stock solution of resveratrol was 50 mM and sucrose was 500 mM.

METHODS:

- Include the molar concentration of all the chemicals used, the way the methods are presented, not possible reproducibility.

Response;- We have included molar concentration as far as possible.

- Amylosucrase DgAS: Please include more details, temperature, pH, molar ratio, ionic strength.

 Response;- We have mentioned the temperature and pH of the experiments. But not about the ionic strength though it was a nice suggestion.

- RESULTS AND DISCUSSION:

- The influence of substrate systems to Amylosucrase DgAS stability was also investigated? The Amylosucrase DgAS showed how about stability?

 Response;- As per the suggestion, amylosucrase obtained from Dinococcus geothermalis (DgAS) exist in a dimer form, so stable against solvent, temperature and other physical conditions. 

- The thermal stability to Amylosucrase DgAS prepared is one of the most important application criteria for diferent applications. This stability depends to Amylosucrase DgAS preparation strategy. It also depends on the stabilization of the Amylosucrase DgAS. This discussion could be improved. Please include in the manuscript.

 Response;- For this, we have addressed it in the discussion section. Also, it has been mentioned in Tian et al, 2018.

- The stability in organic solvents, metal ions, or detergent enables its wide application in synthesis processes which nowadays are in great demand from the point of view of industrial. The effect of organic solvents on the Amylosucrase DgAS activity was studied? For example, in the presence of organic solvents?

Response;- - We appreciate your suggestions to make the manuscript better but we did not perform the experiments as suggested. However, Jang and the team reported that more than 30% of DMSO decreased the production of glucosylated products by 50%. (Jang et al 2018).

- Was determined the full loading of Amylosucrase DgAS prepared under the optimized conditions? This information must be clear in the manuscript.

 Response;- As per the suggestions, we did not determine the full loading of Amylosucrase, DgAS under the optimized conditions of experiments due to time constrain.

- The Amylosucrase DgAS may experience protein aggregation (mainly near to the isoelectric point). This may be caused by undesired Amylosucrase DgAS - interactions where inactivation that can stabilize incorrect Amylosucrase DgAS structures.

 Response;- Yes, the DgAS may suffer from protein aggregation near its isoelectric point.

- The optimization of Amylosucrase DgAS preparation process, the preparations shown having diffusion limitations? Considering the strategy presented in this manuscript. Please, this should be explained in the manuscript. What were the optimum conditions?

 Response;- We greatly appreciate your suggestion but we did not perform immobilization experiments implying different strategies for the best-optimized conditions of immobilization. We will take note to apply the suggestion in the next similar work.

- Effect of solution pH since the solution pH affects the generation of hydroxyl radicals and also influences the surfasse charge and interface potential properties of the catalyst, it is one of the important factors. Amylosucrase DgAS showed considerable improves in the kinetic parameters in terms of activity, specific activity, Km and Vmax, optimum pH and Temperature?

Response;- Thank you so much for the nice suggestion, but we did not perform kinetic studies as we used well-characterized DgAS for our study and focused on the cosmetic values of resveratrol-O-glucosides. But, some papers reported Kinetic studies of Amylosucrase. Van der Veen et al. 2006 and Rha et al.2020.

- Reusability of Amylosucrase DgAS:  The reusability of Amylosucrase DgAS particles is essential while considering reactions. Amylosucrase DgAS reusability was accounted for? Reusability studies showed that the remaining Amylosucrase DgAS assay was obtained to reduce the increasing number of re-use cycles. The reusability of Amylosucrase DgAS without alteration in its load capacity of performance with the resulting is an advantage. After cycles, please, explain these results. What other factors can influence the results achieved? In addition, the results should be compared with other works of literature in the same application line.

 Response;- By immobilizing the DgAS enzyme, DgAS can be used 50 times for the reaction and found 35 times more effective which results in enhancing the efficiency of the glycosylation reaction. Our labmate, Hun Sang Lee did immobilization work with DgAS. (Hun Sang Lee et al 2018).

- Enhanced stability of Amylosucrase DgAS: Other factors that cause the loss of durability and stability should be explained in the manuscript.

 Response;- For this part of the suggestion, we have mentioned it in the discussion section with red text.

Reviewer 3 Report

A nice publication easy to read and understand. Indeed the stability of resveratrol and its bioavailability limits its use. Your works shows that a glycosylated resveratrol is more stable. This had also been shown with dimers such as viniferins. However glycosylation limits the biological activity of the product compared to resveratrol. (cf anti inflammation and anti aging.)

Related to anti aging your test is light and would have deserved more testing.

Line 523 to 528 it is not clear to understand if it is interesting to reduce the activity of the tyrosinase because melanin is a compound which protects the skin against UV. 

For depigmentation or lightening effect I agree that it could be interesting.

So I would like if you can clarify this part. 

Apart from these small points your publication is interesting.

Concerning some little mistakes :

Line 349 there are many maltodextrin

Line 514 in discussion chapter it is however not howefver

Author Response

A nice publication easy to read and understand. Indeed the stability of resveratrol and its bioavailability limits its use. Your works shows that a glycosylated resveratrol is more stable. This had also been shown with dimers such as viniferins. However glycosylation limits the biological activity of the product compared to resveratrol. (cf anti inflammation and anti aging.)

Related to anti aging your test is light and would have deserved more testing.

Response;- Thank you so much for the suggestion, we will focus on it in future similar types of work.

Line 523 to 528 it is not clear to understand if it is interesting to reduce the activity of the tyrosinase because melanin is a compound which protects the skin against UV.

Response;- For this, we have kept the paragraph as it is.

For depigmentation or lightening effect I agree that it could be interesting.

Response;- Thank you for your kind words.

So I would like if you can clarify this part.  

Apart from these small points your publication is interesting.

Concerning some little mistakes :

Line 349 there are many maltodextrin

Response;- As per the suggestion many maltodextrin words are replaced by different maltodextrins.

Line 514 in discussion chapter it is however not however

Response;- The word has changed to however.